# Exploring turn demands of an English Premier League team across league and knockout competitions over a full season

**Erin Griffiths**[1,2]*, **Thomas Dos'Santos**[2], **Christopher Gaffney**[1], **Timothy Barry**[1]

1 Lancaster Medical School, Health Innovation One, Sir John Fisher Drive, Lancaster University, Lancaster, United Kingdom, 2 Department of Sport and Exercise Sciences, Manchester Institute of Sport, Manchester Metropolitan University, Manchester, United Kingdom

* erin.griffiths@stu.mmu.ac.uk

## Abstract

Turns are key performance actions in soccer, but can also induce high mechanical loads resulting in tissue damage or injury. This study aimed to quantify the turn demands of an elite English Premier League soccer team. Turning data were obtained from 49 soccer matches (2022–23 season), from a single team that played 35 Premier League, 5 UEFA Europa League, 5 League Cup and 4 FA Cup matches using Sportlight LiDAR technology. Turns were analysed from 29 players who were categorised in playing position groups: goalkeeper (GK), central defenders (CD), full-backs (FB), central-midfielders (CM), wide-midfielders (WM), central-forwards (CF). Turn categories: high (120–180°), medium (60–119°) and low (20–60°) angled, and very high (>7.0ms⁻¹), high (5.5–7.0ms⁻¹), medium (3.0–5.5ms⁻¹), and low (<3.0ms⁻¹) entry speed (ES) was analysed. Primary findings show, on average, per match, CM performed more total turns (~35), than all other playing positions. Additionally, CM performed significantly more low and medium entry speed and high angled turns than other outfield positions. There were no significant differences between turn frequencies and turn characteristics in different competitions ($p > 0.05$). The turning demands of soccer appear to vary significantly between player position. These findings may help inform position-specific return-to-play protocols, physical preparation strategies, drill design and rehabilitation programmes.

## Introduction

Soccer is a complex sport including prolonged high-intensity intermittent phases of play, rapid changes in velocity and direction, and unpredictable movement patterns [1–3]. The physical demands of soccer are often quantified using running metrics [4], such as distances and frequencies of sprinting, high speed running, acceleration, and deceleration. Turning movements (the term 'turning' is synonymous with change of direction [COD] [5–7]) are essential movements for success in soccer performance [8,9], but can also create tissue damage, and fatigue. Linear demands such as high-speed and sprint running distances ranged from 618 to 1,001 m and 153–295 m, respectively, in professional male soccer players [10]. Conversely, turns have been reported at a higher variability, with recent literature reporting

**Data availability statement:** All raw data files are available from the OSF database (accession number(s) https://osf.io/jz7yc/files/osfstorage/676eb3b02fd8273323c550f5). Please download as a zip to view the file, due to file size.

**Funding:** The author(s) received no specific funding for this work.

**Competing interests:** The authors have declared that no competing interests exist.

turns counts ranging from ~700 [11] to ~38 [7]. These turning metrics are often overlooked and underreported due to some of the challenges of accurately quantifying turning with current technology (i.e., global positioning systems (GPS)).

Only a limited number of studies have quantified turns during soccer matches, but these have been traditionally limited to notational analysis, and generally fail to report the entry speeds and accurately quantify turn angle[5,11]. Bloomfield, Polman and O'Donoghue [5] and Morgan et al. [11] completed a positional based turn analysis for defenders, midfielders and strikers and found ~700, ~500 and ~600 completed turns per match [5], compared to ~299(CD), ~336(CM), ~304(CF) [11], respectively. It is important to note that both studies were inclusive of 0–90° turns which were the most recorded turns (80%[5] vs. 77%[11]). The biomechanical demands of these turns on the body are angle and velocity dependant[7,12–16]. Therefore, quantifying the number of turns alone lacks provides limited practical application without the added context of entry speed and turn angle. Both factors influence the braking and propulsive ground reaction characteristics and external joint moments during the 'plant phase' (the final foot contact when changing direction), in turn, altering the biomechanical demands of the movement and therefore the injury risk and lower limb loading [8,17–20].

Turn analysis studies are often restricted to analysing low entry speed turns [12,14]. To our best knowledge, Dos'Santos *et al.* [7] is the only study to objectively quantify turn associated metrics in elite Premier League soccer uisng LiDAR technology, observing that wide players (WM, FB) performed over 8% of all turns above the $5.55\,\mathrm{ms^{-1}}$ threshold (high entry speed (ES)); these higher entry speed turns are likely to be of more interest to practitioners due to their potential to generate greater knee joint loading, fatigue, tissue damage, as well as successful performance outcomes such as goal scoring [5,7,21]. Greater turn entry speed has demonstrated an increase in the level of deceleration required to change direction; rapid decelerations during these type of movements have been shown to elevate muscle damage and mechanical stress through eccentric muscle actions as well as increase injury risk [8,13,17]. Understanding the frequency of each turn type (based upon entry speed and turn angle category), will be imperative when determining injury risk and lower-limb loading factors for players on both a match-by-match and per-season basis.

Player tracking technology, from a sports science perspective, is used predominantly for external-load monitoring during training and match play, in an effort to reduce injury incidence rates, monitor potential fatigue, and optimize performance [22]. Regular use of this monitoring technology allows for session-to-session adjustment of training to ensure that the training loads physically prepare players for the demands of a match [23,24]. The current industry-dominating tracking systems in soccer is the global positioning system (GPS) [25]. Since the in-competition legalization of GPS technology in 2015 by the International Football Association Board, it has been used by practitioners world-wide to provide on-pitch, external load metrics [25]. It boasts benefits such as its relative affordability, its portable nature, and an instant feedback system; allowing for constant monitoring during both training and matches [23,26]. These GPS systems, when evaluated against the gold-standard 3D-motion capture system, yielded lower error values than video-based systems, but lower validity than local positioning systems (LPS) [27],as well as often failing to quantify turn metrics [26]. However, with the recent integration of LiDAR technology, some of the current limitations faced by GPS, for example, satellite signals being blocked by stadiums/buildings, negative correlation between number of satellites signalling to the receiver and total distance and velocity measurement error, and the wearable nature of the device, could lead to reduced usage in elite soccer. However, a validity study directly comparing LiDAR and GPS needs to be completed before any conclusions can be drawn from these assumptions.

Building on the recent work of Dos'Santos et al. [7], this study aimed to use Sportlight's® LiDAR technology to determine the turn demands experienced by soccer players in an elite male professional team competing in the Premier League, UEFA Europa League, FA Cup and League Cup. Specifically, the study sought to analyse turn characteristics for each playing position (goalkeeper, central defender, full-back, central midfielder, wide forward, and center forward). Additionally, this study aimed to examine differences in turn demands between competitions (Premier League, FA Cup, League Cup, and UEFA Europa League), for a single team. The findings from this study will provide normative data pertaining to the turning demands of elite male soccer players where data is lacking, which practitioners can use to inform injury-prevention, rehabilitation, and return-to-play programs.

## Methods

### Research design

This study used a longitudinal within- and between-subject comparative design, whereby turn metrics were compared between different soccer competition formats and between different soccer players' positional groups in an English Premier League soccer team.

### Game analysis and player data

Turning data were obtained from 49 match fixtures during 2022–23 season, from the Premier League (35 matches), UEFA Europa League (5 matches), League Cup (5 matches) and FA Cup (4 matches), the latter three competitions are grouped for analysis as 'knockout' competitions. All turn data were collected using Sportlight®'s LiDAR tracking system (Sportlight®, Oxford, UK; LiDAR). For home matches, the single-sensor system was permanently mounted 7 metres above pitch height, sampling at 1.2million spatial readings per second over a 200m range at 10Hz. Data was collected for away games where Sportlight® was installed. The installation position of each unit in away stadiums varied.

The proprietary software utilized in conjunction with the LiDAR system facilitated the tracking of all movements occurring on the pitch. Clusters of moving points were detected to pinpoint the positions of players in the foreground plane. The software then determined the centre of each cluster, a method proven to yield accurate positional data when compared to a 3D motion capture system utilizing a four-marker pelvis model [26,28]. To ensure each cluster point was continually allocated to the same player the LiDAR system worked in conjunction with three cameras which captured high-resolution imagery (Sony IMX253, 12.4MPx, 10fps synchronized with the LiDAR data). Their output was fed into an artificial intelligence system, which undertook the temporal tracking of individual clusters and the re-identification of players using previously captured imagery. Gait-neutralized velocity was computed by subjecting the raw velocity data to a fourth-order low-pass Butterworth filter (with a 1 Hz cut-off [26]). Gait-neutralized acceleration was defined as the alteration in gait-neutralized velocity over time [7,26]. The Sportlight® system's output provided both positional data and derived metrics for turning.

Overall, data from 29 (age 27.3 ± 4.3 yrs; height 183.7 ± 6.7 cm; mass 74.7 ± 6.7 kg) different players were analysed. All players were grouped into similar positional group similar to previous research [7,29]: goalkeeper (GK: n = 3), central defender (CD: n = 4), full-back (FB: n = 5), central midfielder (CM: n = 8), winger forward (WF: n = 6), central forward (CF: n = 3). Variations from previous research are found in using the term 'winger forward' instead of 'winger midfielder' as this is a more applicable term for the wide players considering the teams formation. GK's were also analysed as limited research has been conducted on turns completed in this position. Through freely available online

information (https://www.footballcritic.com/), formation was determined to be predominantly 4-2-3-1 (42 matches), other formations included: 4-3-3 (5 matches) and 4-1-4-1 (2 matches). If players had played in multiple positions, they were categorized into the position group in which they had played most matches. All players played >80% of their matches in their allocated position group. Analysis which considered per-match averages only included turn data from players who played >85 minutes as this is considered a 'full-session' by Sportlight®. Analysis which looked at individual turn characteristics, i.e., average entry speed (ms⁻¹) for each turn category, considered all turns, regardless of the athletes playing time. Ethical approval was granted by the university institutional review board (ID: LMS-23–1-Griffiths), the soccer team provided informed written consent to publish the data for research purposes. All data were received by authors for research purposes on the 19th of June, 2023.

## Turning metrics

Turn data were only collected and analysed if the turns were considered 'significant' [7] as per manufacturer algorithm. A significant turn is postulated to elicit high biomechanical loads and emits the inclusion of lower biomechanical load change of direction movements such as curvilinear running. Specific metrics which qualify a turn to be 'significant' are as follows: a change of direction with a deceleration <-2 m/s² upon entry to the turn, an angle change in direction of travel ≥20°, and a subsequent acceleration ≥2 m/s² whilst exiting the turn, all within a 1 second duration. Turns were further sub-categorised by angle and entry speed (Table 1) [7].

Unpublished observations highlighted mean absolute errors of <2.2% for Sportlight® turn angle and entry speed compared to the gold standard of 3D motion capture (Qualisys AB, Gothenburg, Sweden, v2021.1.2). For football-specific movements (i.e., jogging, linear

**Table 1. Definitions of entry speed (ms⁻¹), turn angles (°) and their sub-categorisations. Frequency of total turns and entry speed/turn angle sub-categories were calculated.**

| Sub-Category/Turn Characteristic | Definition |
|---|---|
| Low Entry Speed | When entering the turn, the instantaneous speed of the player upon initiation of the deceleration (when the players deceleration exceeded ≤−2 m/s²) is considered to be 'low'. (<3.0ms⁻¹) |
| Medium Entry Speed | When entering the turn, the instantaneous speed of the player upon initiation of the deceleration (when the players deceleration exceeded ≤−2 m/s2) is considered to be 'medium'. (3.0–5.5ms-1) |
| High Entry Speed | When entering the turn, the instantaneous speed of the player upon initiation of the deceleration (when the players deceleration exceeded ≤−2 m/s2) is considered to be 'high'. (5.5–7.0 ms-1) |
| Very High Entry Speed | When entering the turn, the instantaneous speed of the player upon initiation of the deceleration (when the players deceleration exceeded ≤−2 m/s2) is considered to be 'very high'. (>7.0 ms-1) |
| Low Angle | The angle of the turn (see definition below) was 20–59° |
| Medium Angle | The angle of the turn (see definition below) was 60–119° |
| High Angle | The angle of the turn (see definition below) was 120–180° |
| Entry Speed | The instantaneous speed of the player upon initiation of the deceleration (when the players deceleration exceeded ≤−2 m/s²) (17) |
| Turn Angle | The angle of the turn was computed as the angle between the acceleration and deceleration vector in the horizontal plane based on the estimated whole-body centre of mass (estimations were based on cluster points used to localize players by proprietary software)(17,20) |

sprinting, curved sprinting, and turns of various angles), the system has been validated against gold standard three-dimensional motion capture systems [26]. Trivial to small differences in time spent during different speed zones (d = 0.04–0.26) were observed, and root mean square error values of 0.04–0.014 m/s and 0.16–0.7 m/s$^2$ for velocity and acceleration data, respectively.

## Statistical analysis

All statistical analyses were performed using the coding software R (version 2023.06.1+524). Normality was assessed using a Kolmogorov-Smirnov test. For variables with >5000 data points, a density graph was used to visually determine normality; analysis confirmed all tests were non-parametric. Positional differences of turn characteristics and frequencies were determined via Kruskal-Wallis tests. Differences in turn characteristics between competitions were calculated using a Wilcoxon Signed-Rank test. Statistical significance was defined as $p<0.05$ for all tests. In the event of a significant result, the Dunn test (1964) pairwise comparison, adjusted via the Holm correction, was applied. Epsilon squared effect sizes were also calculated for all Kruskal-Wallis tests; 0.01–0.059 (small effect), 0.06–0.139 (moderate effect) and 0.14 (large effect) [30, 31] presented for positional and competition comparisons reflect the average turning frequency per match. Using Dos'Santos et al., (2022) effect size (0.19) a calculation was completed on G*Power (p=0.05, power=0.8). Results from these calculations concluded a minimum of 40 match observations were required for positional group comparison. As such, the 49 matches analysed satisfies our calculations.

The interquartile range (IQR) was calculated by subtracting the first quartile (25th percentile) from the third quartile (75th percentile) of the dataset. A polar scatter chart was used to display turn incidence rates, across the entire season.

## Results

### Total number of turns per position

The overall average number of turns per player per match was 24.5. Position-specific averages can be found in Fig 1 and S3 Table. CMs performed significantly more turns on average per match than WF and GK ($H_5 = 154.25$, $p < 0.05$).

### Turns by angle and position

No significant differences were found between outfield players for low angled turns ($p > 0.05$) (Fig 2d). CMs performed significantly greater medium angled turns frequencies than all position groups excluding CF($p<0.05$). CMs and FBs performed significantly more high angled turns than GK, CD and WF ($p<0.05$). Differences between overall means, medians and IQR for each turn angle category are displayed in a table as S3 Table.

### Turns by entry speed and position

CM performed more low entry speed turns than FB and WF ($p <0.05$); CD performed significantly more low entry speed turns than WF ($p <0.05$) (Fig 2a). Significant differences were also identified between positions for medium entry speed turns ($H_5 = 135.54$, $p < 0.01$), with CM performing more than CD, FB and WF ($p <0.05$) and FB completed significantly more than WF ($p <0.05$) (Fig 2b). CD completed significantly less high entry speed turns on average per match than CM, FB, and WF ($H_5 = 34.43$, $p < 0.01$)(Fig 2c). Overall mean±$SD$, median and IQR for each turn entry speed and position are displayed as a table in S4 Table.

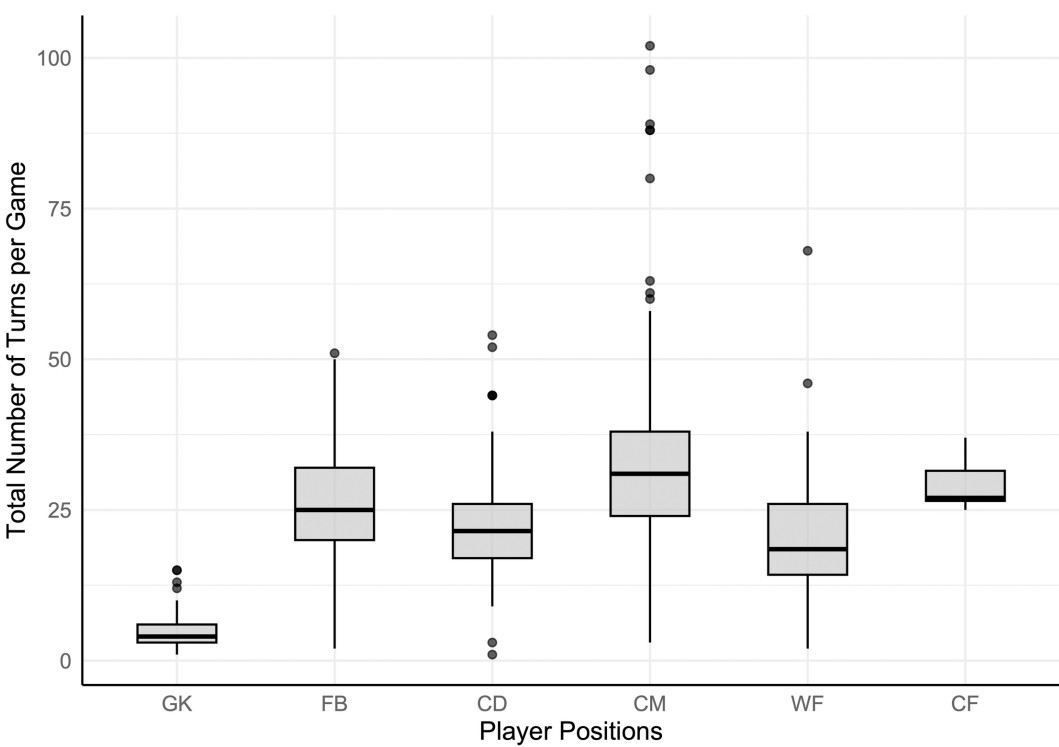

**Fig 1. Total turn count for each player position, per game.** The interquartile range (IQR; length of the box), median (intersecting line within the box), range (solid line) and outliers (singular points) of the total number of turns per game are displayed.

### Proportions of turns completed in each entry speed and angle group per position

Similarities were found between player position groups for the proportions of turns completed in each entry speed groups, despite a large range in sample size (224–5693): GK: $X^2$ (2, 224) = 102.38, $p$ =.001; FB: $X^2$ (2, 2862) = 2229, $p$ =.001; CD: $X^2$ (3, 2041) = 1741, $p$ =.001; CM: $X^2$ (3, 5693) = 5169.5, $p$ =.001; WF: $X^2$ (3, 2778) = 2125.6, $p$ =.001; CF: $X^2$ (3, 709)= 570.13, $p$ =.001. Visual demonstrations of the above data can be found in S1b Fig.

Proportional turn angle incidence rate similarities were found between position groups, despite a large range in sample size (224–5693): GK: $X^2$ (2, 224) = 165.06, $p$ =.000; FB: $X^2$ (2, 2862) = 1615.7, $p$ =.001; CD: $X^2$ (2, 2041) = 989.11, $p$ =.001; CM: $X^2$ (2, 5693) = 3169.5, $p$ =.001; WF: $X^2$ (2, 2778) = 1655.9, $p$ =.001; CF: $X^2$ (2, 709)= 319.97, $p$ =.001. Visual demonstrations of the above data can be found in S1b Fig.

### Investigating how entry speed changes with turn angle

Mean entry speed for high angle turns was 3.48ms$^{-1}$, medium angle turns 3.63ms$^{-1}$, and low angle turns 3.66ms$^{-1}$. High angled turns were found to elicit significantly lower entry speeds than medium and low angled turns ($H_2$ = 56.10, $p$ < 0.01). The small effect size must be noted, $e^2[H]$ = 0.004. The difference between means can be seen in S4 Table. The IQR, outliers, range and median entry speed for each angle group are demonstrated in Fig 3d.

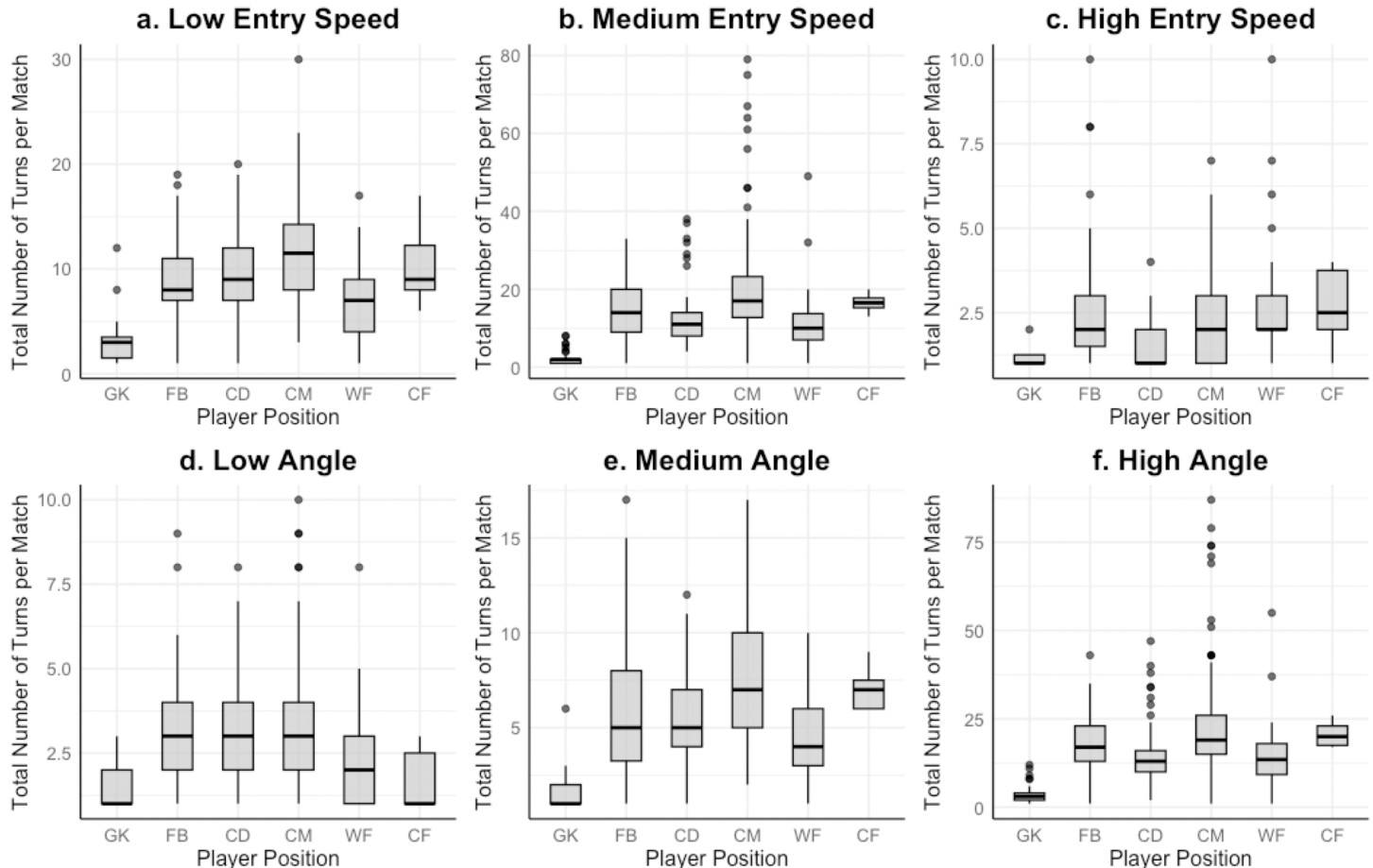

**Fig 2. Total turn count per match for each position, for entry speed and turn angle groups.** The interquartile range (IQR; length of the box), median (intersecting line within the box), range (solid line) and outliers (singular points) of low entry speed turns (Fig 2a), medium entry speed turns (Fig 2b), high entry speed turns (Fig 2c), low angled turns (Fig 2d), medium angled turns (Fig 2e) and high angled turns (Fig 2f) are displayed. Results for very high entry speed turns are not displayed in Fig 2 due to the lack of data points detected.

### Entry speed variations between positions for each angle group

CD performed significantly slower entry speed high angled turns than FB, CM, WF and CF; GK performed significantly slower entry speed high angled turns than FB, CM, WF and CF ($H_5 = 102.93$, $p < 0.01$). FB, CM, WF and CF completed significantly faster medium angle turns than GK and CD, whilst CM were significantly slower than WF in this turn category ($H_5 = 77.25$, $p < 0.01$). Significant positional differences in entry speed for low angled turns were as followed: CD<CF,FB,WF; WF>GK,CM,CD; GK<FB,CD,CM,WF,CF ($H_5 = 77.25$, $p < 0.01$) The entry speed median, IQR, range and outliers can be found in Fig 3c (high angle), Fig 3b (medium angle) and Fig 3a (low angle) for all playing positions. Overall effect sizes: position group $e^2[H] = 0.014$, and angle group $e^2[H] = 0.003$.

### Entry speed and turn angle incidence rates

High turn incidence rates lie within the boundaries of 125–130° and 3.40–3.60ms⁻¹, this can be viewed in the histograms in the supplementals (S2a Fig and S2b Fig). The polar scatter chart (Fig 4) displays turns via entry speed (radial markers) and turn angle (degrees plotted on

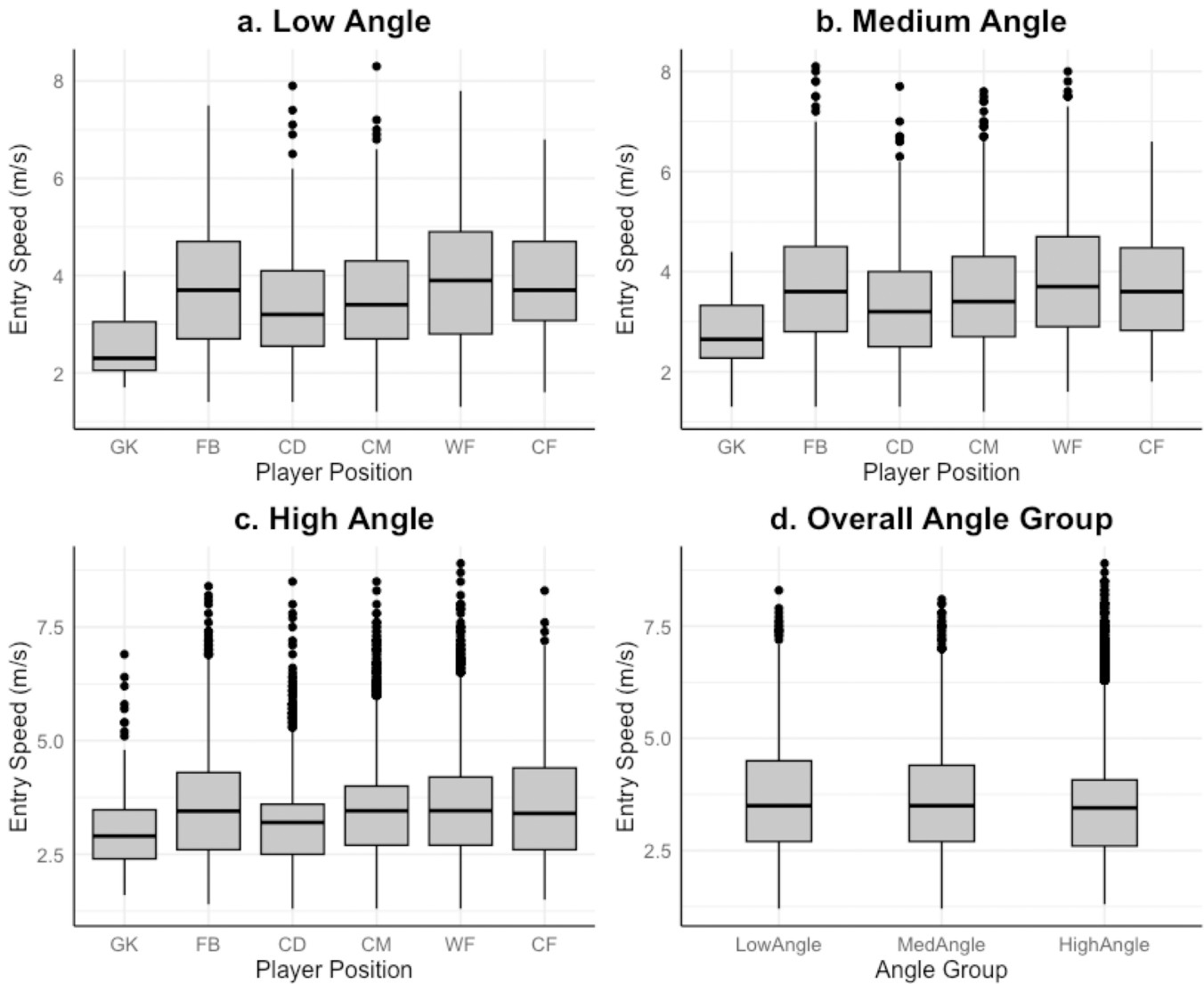

**Fig 3. Entry speed (ms-1) for each angle group.** The interquartile range (IQR; length of the box), median (intersecting line within the box), range (solid line) and outliers (singular points) of Entry Speed (ms-1) for low angle (Fig 3a), medium angle (Fig 3b), high angle (Fig 3c), and overall angle group (Fig 3d), turns are displayed.

circumference) and highlights high occurrence rates of turns at a high angle and low to medium entry speeds (ms-1) and low occurrence rates of turns at low angle/very high and high entry speed.

## Competition differences

There were no significant differences between the number of turns (across all positions) completed per match, as a team, when analysing knockout vs league soccer ($p > 0.05$; $e^2=0.02$ (small)). See Table 2 for turn angle and entry speed specific turn counts.

## Discussion

The present study sought to determine the turn demands of soccer players in each playing position using data from the English Premier League, FA Cup, League Cup and Europa

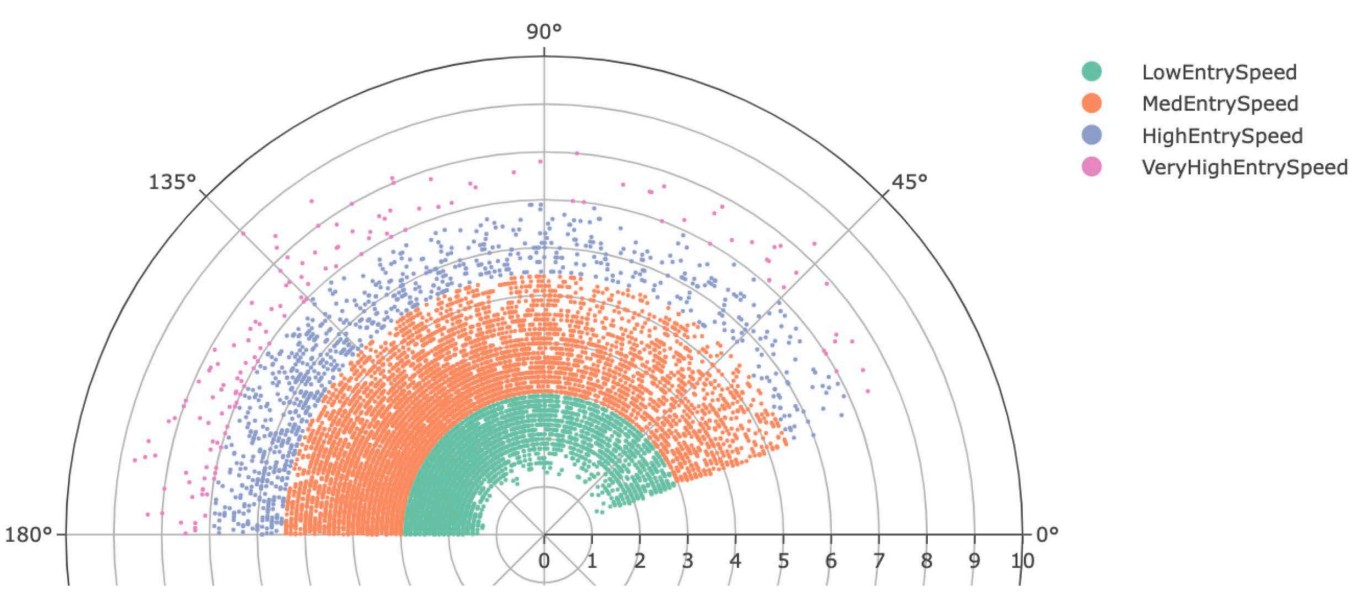

**Fig 4. Distribution of turns demonstrated by their turn angle (θ) and entry speed (radial).**

**Table 2. Identifying the number of turns completed (for each turn characteristic sub-category) per match for players in League vs. Knockout matches. (mean ± SD).**

|  | No. of Turns per League Match | No. of Turns per Knockout Match |
|---|---|---|
| **Total Turns** | 25.10±16.30 | 22.81±11.84 |
| **High Angle** | 17.65±13.27 | 14.58±7.85 |
| **Medium Angle** | 5.65±3.18 | 6.30±3.78 |
| **Low Angle** | 2.97±1.81 | 2.78±1.74 |
| **Very High Entry Speed** | 1.09±0.29 | 1.05±0.23 |
| **High Entry Speed** | 2.39±1.55 | 2.10±1.49 |
| **Medium Entry Speed** | 15.01±12.14 | 12.17±7.12 |
| **Low Entry Speed** | 8.80±4.69 | 9.37±5.14 |

League matches: Premier League. Knockout matches: Europa League, League Cup, FA Cup.

League of a single team. The primary findings showed no difference in turn demands between competition types (Premier League vs. knockout matches), but significantly greater total turns performed by CM (~35) compared to GK (~5), CD (~23) and WF (~21), and differences in turn characteristics between each position group for each angle and entry speed category.

In contrast to the findings of the present study, similar research from Dos'Santos *et al.*[7] showed CMs performed more turns on average (~38) compared to other positional groups (FB, CD, CF). Though CM performed more turns, the rationale for CM performing the most turns in both studies was shared; this was attributed to the duel attacking and defending roles.

Large discrepancies between the current study and other turn analysis literature [5,11] are attributed to the lack of strict criteria applied to the classification of turns, such as a deceleration threshold that needs to be exceeded or no minimum turn angle for analysis. Most turns that occur in soccer are low biomechanical load, near-liner and <90° (~600 of the ~700 turns) [5]; through the exclusion of <20° turns and a strict deceleration and reacceleration threshold, the present study does not quantify and therefore excludes a high number of turns

often included by other researchers. This can be observed in the variance in results with total average turns per player being ~726 [5], ~304 [11], ~183 [32] compared to the current study's finding of ~24. With turn ability discriminating between elite and sub-elite [33], it is likely the large variances between subject groups training status will also increase inconsistencies between findings. Only Bloomfield, Polman and O'Donoghue [5] and Dos'Santos *et al.*[7] have quantified turn frequency in Premier League soccer players. This calls for further turn analysis to be completed on Premier League and other 'top-flight' soccer leagues.

## Turns by angle and position

Practitioners can use the data from this study to design position-specific drills by monitoring the volume of repetitions completed rather than altering the angles within the drill. This is due to all positions demonstrating proportionally the same split (high, medium and low angled turns). However, positional differences in frequency of turns vary between angle groups, such as: CM and FB perform significantly more high angled turns, whereas only CM performed significantly higher medium angled turns. This understanding of positional-based differences of turn angles allows practitioners to tailor their drills for players based on their position to ensure they are prepared for the demands of the match.

Sharper turning movements elicit greater lower-body loading [12,14,8] such as increased impact ground reaction forces and knee adduction moments which can increase tissue loading. These movements, however, are unavoidable in sports therefore it is imperative practitioners ensure athletes can perform these high-angled turns with the correct mechanics [34–38] and have the physical capabilities to endure the knee-loading associated with them [8,39,40]. Therefore, the current findings will help improve the knowledge of the frequencies of these high-angled turns so practitioners can adequately prepare players for the high biomechanical loads and ensure correct execution to reduce injury risk factors.

## Turns by entry speed and position

Increasing entry speed during turning can also increase knee mechanical loading such as peak knee abduction moment [41–43], trunk deceleration [43] and peak posterior ground reaction forces [44]. Interestingly, only 7.7% of all turns were completed at a 'high' or 'very high' entry speed, proportionally, all outfield positions completed similar proportions of these 'high' and 'very high' entry speed turns, ranging from 10.8% (CD's) to 4.7% (FB's). Though CD performed proportionally the largest amount of 'high' and 'very high' turns, they performed significantly less high entry speed turns per match than CM, WF and FB's.

The results of these findings in WF and FB could be attributed to tactical and contextual factors, such as the opportunity to cover greater entry distances through channelling, overlapping, and recovery runs. Teams often use their fastest players in wide positions because these areas need to be exploited in build-up play to achieve a goal, with pace being a crucial mechanism for this exploitation. Abbott, Brickley, and Smeeton [45] found that wide attackers and wide defenders attain significantly higher peak speeds than all other outfield positions, further supporting this argument. As CM performed significantly higher overall turns, likely due to their dual attacking and defending role and therefore their increase in game-involvement, it is unsurprising they have a significantly greater number of 'high' and 'very high' entry speed turns as this is in keeping with the total increase in turns throughout most other categories.

Throughout all outfield positions, medium entry speed turns were performed significantly more than any other entry speed category (~57.7%). Many studies have categorised 'fast' approach speed to turns to be between 4.0–5.0ms⁻¹ [41,43] which falls within the 'medium' category for this study. In the studies where turn entry speed has been analysed, there have

been factors which may explain these lower boundaries. This must be acknowledged when interpreting biomechanical load findings which use these lower boundaries to describe 'fast' turns. Significant increases in lower limb loading and injury risk factors have been identified during turns completed with these 4.0–5.0ms⁻¹ approach speeds, which falls into the category of 'medium entry speed' in the current study [8,41,43]. This is why the findings that significantly more turns occur at a medium entry speed than any other entry speed group is highly important as we know that players are undergoing high biomechanical loads at high frequencies throughout the match, therefore preparation must focus on turns at this speed to ensure accurate physical preparation.

### Entry speed and angle groups

Though Dos'Santos et al. [7] completed a detailed analysis on both turn entry speed and turn angle in English Premier League soccer, there was no combined analysis of the relationship between two variables. The current study aimed to further analyse these key turn characteristics to gain a greater level of understanding of the overall turn demands. S2 Fig demonstrates high angled turns elicit significantly lower turn entry speeds compared to medium and low angled turns. This finding is unsurprising based on the angle-velocity trade-off whereby typically sharper turns require greater acceleration and reduced entry speeds to achieve deflection of the COM components when turning in response to a stimulus [8].

Dos'Santos et al. [7] reported that ~63–70% of all turns were high angled, alongside ~43.3–56.8% of all turns completed at a medium entry speed (3.0–5.5ms⁻¹). The current category boundaries could be considered broad as significant differences in biomechanical loading factors have been found between entry speeds of $3.82 \pm 0.36$ms⁻¹ and $4.82 \pm 0.58$ms⁻¹ [43], as well as turn angles of 90° and 180° [18]. These figures are similar to the boundaries of the most frequented turn categories (medium entry speed and high turn angle). Hence, the current study acknowledged the importance of determining where within these categories the highest incidence rates of turns occurred to ensure correct understanding of biomechanical load effects. S2 Fig combines both turn angle and entry speed to highlight the high incidence rate of turns, which occurred over the season, between 125–130° and 3.40–3.60ms⁻¹. Though this aligns with Dos'Santos et al. [7], further research should be completed using the current turn definition to determine if this hot spot of turn frequency rate is an accurate representation of all 'significant' turns or a limiting factor. For example, it could be hypothesised that turns at high turn angles and medium to very high entry speeds may still elicit very high biomechanical loads but may not be included in analysis due to change of direction time being >1 second. It is understood that increased entry speed corresponds to decreased turn angle deflection [12,32,46], though, factors such as decreased requirements for deceleration [47] may explain their exclusion from this study, rather than their lack of existence within matches.

### Turn demands and competition types

Previously differences in external load intensity had been identified based upon opposition quality [48], game importance [49], and tournament standard (i.e., national vs regional) [49]. Despite these previous findings, the current study found no significant differences between turn characteristics when comparing knockout (FA Cup, League Cup and UEFA Europa League) vs. league soccer (Premier League). Previous research, which has focused on external load intensity across competition types, has never compared turn frequency and associated metrics, therefore it can be concluded that turns do not follow the same pattern as other key performance indicators such as total distance covered, maximum sprint speed and number of accelerations/decelerations which differs between league and knockout football [48,50].

However, the current study is specific to 'significant' turns, therefore, finding a significant difference between already high intensity turns may be unlikely. Equally, research on external load in different competition types has yet to analyse turns, nor has research often studied top-flight elite soccer teams, therefore, previously drawn conclusions which attribute intensity differences to opposition quality [48] and tournament standard [50] may not be applicable to the current study.

## Limitations

It should be noted that limitations within the current study are predominantly due to the novel nature of the technology and research. The turn classification used within the present study has only been used once previously [7], all other research in this area provides different definitions and criterion [5,11,29,32,51,52].

The inclusion of only players who have completed >85 minutes per match, rather than like-for-like substitutions being considered the same observation [7], reduces the participant sample size; CF's contained only 3 soccer players. Future research should consider recording formations and positional swaps during games to ensure substitutions do not result in lost observations. In addition, each player was allocated a position group which was adopted for all matches. Although no players were included in our study that played less than 80% of their matches in their allocated position group, it is important to acknowledge that this is a limitation, and future research should aim for 100% position-specific data. Furthermore, formation changes within games should be considered; the current study recorded only the starting position of the player; therefore, multiple positions may be included within one observation if a player has changed position mid-match.

Practitioners should refrain from generalizing the present study's findings as they are likely to be influenced by the specific team formation, skill level, tactics, style of play, opposition and contextual factors. This highlights the scope for further research to investigate turn demands for each of these variables, as well as age and sex.

## Conclusion

This study provides further and novel insights into the turn demands of an elite Premier League Soccer team who also competed in the FA Cup, League Cup and UEFA Europa League. CMs were found to perform the most turns, and high angled, medium entry speed turns were the most frequently performed. Overall, negligible differences in both turn frequency and turn characteristics were observed between soccer competition formats.

These findings provide normative data pertaining to the turn demands within elite soccer match play, which can be used to help inform return-to-play protocols, physical preparation strategies, drill design and rehabilitation programmes, with emphasis placed on the specific turns which occur the most during matches.

## Supporting information

**S1 Figure. The proportion of all turns completed, in all competitions, for: (1a) each angle group, per position (Low Angle: 20°-60°, Medium Angle: 60.1°-120°, High Angle: 120.1°-180°); (1b) each entry speed group (Low ES: <3.0 ms$^{-1}$, Medium ES: 3.0–5.5 ms$^{-1}$, High ES: 5.5–7.0 ms$^{-1}$, Very High ES >7.0 ms$^{-1}$).** The proportion of turns for each position can be seen as follows: Goal Keepers; Full Backs; Central Defenders; Central Midfielders; Winger Forwards; Central Forward.
(TIF)

**S2 Figure. Cumulative turns from across all competitions, this figure highlights the increased frequencies within specific angle (2a) and entry speed (2b) parameters.** (TIF)

**S3 Table. Identifying the number of turns completed (for each angle category) per match for players in each position group(mean ± SD).** Inter-quartile range (IQR). 95% Confidence Intervals. *GK: Goalkeeper. FB: Full-back. CD: Central Defender. CM: Central Midfielder. WF: Winger Midfielder. CF: Central Forward.* (DOCX)

**S4 Table. Identifying the number of turns completed (for each entry speed category) per match for players in each position group (mean ± standard deviation (SD)).** Inter-quartile range (IQR). *GK: Goalkeeper. FB: Full-back. CD: Central Defender. CM: Central Midfielder. WF: Winger Midfielder. CF: Central Forward.* (DOCX)

## Author contributions

**Conceptualization:** Erin Griffiths, Timothy Barry.

**Formal analysis:** Erin Griffiths.

**Investigation:** Erin Griffiths.

**Methodology:** Erin Griffiths, Timothy Barry.

**Project administration:** Erin Griffiths, Christopher Gaffney.

**Supervision:** Christopher Gaffney, Timothy Barry.

**Writing – original draft:** Erin Griffiths.

**Writing – review & editing:** Erin Griffiths, Thomas Dos'Santos, Christopher Gaffney, Timothy Barry.

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
