## [Decision Letter · Decision Letter 0]

22 Oct 2024

PONE-D-24-31277An Exploratory Analysis Investigating the Significant Turn Demands of the Premier League, FA Cup, League Cup and UEFA Europa League for an English Premier League Soccer TeamPLOS ONE

Dear Dr. Griffiths,

Thank you for submitting your manuscript to PLOS ONE. After careful consideration, we feel that it has merit but does not fully meet PLOS ONE’s publication criteria as it currently stands. Therefore, we invite you to submit a revised version of the manuscript that addresses the points raised during the review process. Please consider the suggestions of both reviewers, namely in methods.

We look forward to receiving your revised manuscript.

Kind regards,

Filipe Manuel Clemente, PhD

Academic Editor

PLOS ONE

Journal Requirements:

3. Please upload a copy of Figure 6, to which you refer in your text on page 25. If the figure is no longer to be included as part of the submission please remove all reference to it within the text.

4. We note you have included a table to which you do not refer in the text of your manuscript. Please ensure that you refer to Table 4 in your text; if accepted, production will need this reference to link the reader to the Table.

5. We notice that your supplementary figures are uploaded with the file type 'Figure'. Please amend the file type to 'Supporting Information'. Please ensure that each Supporting Information file has a legend listed in the manuscript after the references list.

Reviewers' comments:

Reviewer's Responses to Questions

**Comments to the Author**

1. Is the manuscript technically sound, and do the data support the conclusions?

Reviewer #1: Partly

Reviewer #2: No

2. Has the statistical analysis been performed appropriately and rigorously? 

Reviewer #1: Yes

Reviewer #2: Yes

3. Have the authors made all data underlying the findings in their manuscript fully available?

Reviewer #1: Yes

Reviewer #2: Yes

4. Is the manuscript presented in an intelligible fashion and written in standard English?

Reviewer #1: No

Reviewer #2: Yes

5. Review Comments to the Author

Reviewer #1: Dear Authors,

The study design is interesting and align with the scope of PlosOne, I believe it can provide useful information; however, different sections of the manuscript need to be corrected to a large extent.d. I will outline below point by point some suggestions for improvement, but the main issues that need to be addressed are as follows:

• Results section: It is disorganized, data is repeatedly presented in both tables and figures, and the significance levels are not included in the tables and figures.

• Stated objectives: The necessary methodology to address all of the objectives has not been applied, and the objectives themselves are not properly written.

• Writing of the discussion.

Point by point:

Why do the authors refer to it as "COD" throughout the text, but use "Turn" in the title? I understand that this may be to avoid repetition of keywords, but it might be more appropriate to switch them.

Why does the title include the term “significant”?

Introduction Section

In the introduction, some aspects need to be more thoroughly justified. The authors conclude by stating they will use Sportlight’s® LiDAR technology and provide only a justification for its use, but this approach is not appropriate for the introduction. Instead of saying what they will use, the authors should describe the technologies available for measuring this aspect, their characteristics, and what factors need to be considered. The introduction is meant to establish context and provide a theoretical framework, not to describe the specific methodology that the authors will use.

Additionally, I believe it is necessary to delve deeper into the quantitative demands of soccer, specifically in relation to COD.

Line 55: This is the first time the acronym "COD" appears.

Lines 65-67: Please use the acronym correctly.

Line 61: The terms "Turn" and "COD" seem to be used interchangeably. Are they meant to refer to the same thing?

Line 67: What is meant by “absence”?

Lines 93-100: The objectives need to be revised and not listed as bullet points. Additionally, the authors are not analyzing the demands across different competitions, but rather within one team in the same competition. The last objective cannot be answered with the proposed methodology and is not addressed throughout the manuscript.

Methods Section

Lines 108-131: Can the authors include values for the validity, reliability, and reproducibility of the methodology? The description currently only outlines the process but does not report its validity.

Lines 143-144: I do not understand the following statement: “If players had played in multiple positions, they were categorized into the position group in which they had played most matches.” Wouldn't it be more appropriate to categorize them based on the position they played in each match?

Line 144: The phrase "29 players played 80% of the matches" is unclear. Does this mean "No player had played less than 80% of their games in their allocated position group"?

Line 146: Please include a reference to justify that 85 minutes is considered a full session. Previous studies have used fewer minutes.

Definition of significant turns: This needs to be more clearly and precisely defined.

Statistical analysis: Please explain the process of determining the interquartile range.

Results Section

Table 2 Title: There is an unnecessary period.

Table 2: I believe the table could be formatted differently to improve comprehension. Instead of listing data sequentially, perhaps several values could be included within the same cell (in parentheses)? Why report both the mean and the median?

Line 124: Should this reference be to Table 2 or Table 3?

What is the difference between Table 1 and Figure 1, apart from expressing the same data in different formats? In my opinion, this is redundant. Perhaps only the figures should be included. Please also consider my earlier comments regarding Table 2.

Line 203: The authors state “No significant differences were found between outfield players (p > 0.05) (Fig 2d),” but then report differences between positions.

The Results section needs a complete revision. It is not well organized, as the authors first present all the tables and figures, then repeat each one. Additionally, figure captions appear within the text, the narrative jumps from one table to another, and then back again.

Lines 260-261: I believe the heading is not appropriate for the content that follows.

Why are there no indicators (simbols) in the tables and figures to help interpret significant differences?

Discussion Section

Lines 308-318: The writing and composition of the first paragraph need improvement. Follow the order of the objectives when reporting the main findings. It is not correct to state, “The aim was to build upon research by Dos’Santos et al. (17)”—the objectives are different. Avoid referring to tables and figures. Instead, state the key contributions of the study.

Line 320: If the authors mention there are large discrepancies, they need to report those discrepancies clearly, providing their own quantitative values and those of the references they are comparing against.

Lines 338-340: The authors merely restate the results without explaining, debating, or discussing them. This needs to be expanded upon.

Avoid referring to tables and figures in the discussion section by merely restating previously mentioned results.

Line 365: Please review the format of the final citation.

Lines 421-426: Include references.

What are the practical applications of this study? Could practitioners use these results to create a player profile

Reviewer #2: An Exploratory Analysis Investigating the Significant Turn Demands of the Premier League, FA Cup, League Cup and UEFA Europa League for an English Premier League Soccer Team.

Thank you for the opportunity to review this manuscript, which aimed aimed to investigate the relationship of aerobic capacity to spinal curvature and mobility in young soccer players. In this sense, it appears that the study design was well-thought out and accurately replicates what could conceivably be implemented in practice. In addition, the paper mentioned an important topic that provides in handball referees. However, I have certain doubts in the introduction, the design and also in results and discussion.

Abstract

The abstract succinctly summarizes the study's objectives, methods, and key findings. It effectively communicates the significance of change of direction (COD) movements in soccer and their potential implications for player performance and injury risk.

1. The abstract should be indicated at the end as well as in the conclusions section some directions about these retrospective and perspectives futures.

2. The results need to be better explained.

Introduction

The introduction provides a solid background on the importance of COD in soccer, citing relevant literature to establish the context. It outlines the research gap the study aims to address, specifically the lack of comprehensive data on turning demands across different competitions and player positions.

1. More information is needed to understand the background of the research topic. The introduction must be rewritten, since it is insufficient

2. The literature must be updated an also take into account the last research carried out in the last five years.

3. Please improve the last paragraph and connect the background section with the relation with the aims at the end.

Methods

The study employs a longitudinal within- and between-subject comparative design, which is appropriate for analyzing turn metrics across different competitions and player positions. The use of Sportlight®’s LiDAR tracking system for data collection is a strength, as it allows for high-resolution tracking of player movements. The methodology is well-detailed, including the setup for home and away matches, which enhances the reproducibility of the study.

1. Sample. Can you add in method section a paragraph with the sample size calculated with G-power? Can you add information that have relationship with studies utilized in the introduction. Include Sample size calculation with G*Power (www.gpower.hhu.de

2. Where is the inclusion or exclusion criteria? Incomplete

3. The table 1. The The definition of each subcategory must be explained much better, and the values given in other columns

Result

The results are presented clearly, with a focus on the average number of turns performed by different player positions. The findings indicate that central midfielders (CM) perform significantly more turns compared to other positions, which is a valuable insight for coaches and trainers. The categorization of turns based on angle and entry speed provides a nuanced understanding of player movements.

1. Please follow the instructions for the review when presenting tables and figures.

2. Add information about 95% CI, upper and lower.

Discussion

The discussion effectively interprets the results in the context of existing literature. It highlights the implications for position-specific training and rehabilitation programs, emphasizing the need for tailored approaches based on the unique demands of each position. The authors also acknowledge the limitations of the study, such as the focus on a single team, which could affect the generalizability of the findings.

Conclusion

The conclusion summarizes the key findings and their relevance to soccer training and injury prevention. It reinforces the importance of understanding turn demands to inform training strategies and return-to-play protocols.

The article is well-structured and provides valuable insights into the turning demands of elite soccer players. The use of advanced tracking technology and a robust methodological approach strengthens the findings. However, future research could expand on this work by including a larger sample size and examining additional contextual factors such as player age and sex.

References

The article cites relevant studies and literature, which supports the claims made throughout the text. This enhances the credibility of the research and situates it within the broader field of sports science.

In summary, this study contributes significantly to our understanding of the physical demands placed on soccer players during matches and offers practical applications for training and rehabilitation.

6. PLOS authors have the option to publish the peer review history of their article (what does this mean? ). If published, this will include your full peer review and any attached files.

**Do you want your identity to be public for this peer review?** For information about this choice, including consent withdrawal, please see our Privacy Policy .

Reviewer #1: **Yes: ** Alejandro Rodríguez Fernández

Reviewer #2: No

---

## [Author Response · Author response to Decision Letter 0]

27 Dec 2024

We would like to thank the reviewers for their comments on our work, which provided suggestions that have further improved the manuscript. We have responded to each of the reviewer comments below and where we have made a careful revision of the manuscript, these changes are tracked changes. If you have any questions regarding our rebuttal or revisions, please do not hesitate to contact us.

Journal Requirements:

#1: Please ensure that your manuscript meets PLOS ONE's style requirements, including those for file naming. The PLOS ONE style templates can be found at https://journals.plos.org/plosone/s/file?id=wjVg/PLOSOne_formatting_sample_main_body.pdf and https://journals.plos.org/plosone/s/file?id=ba62/PLOSOne_formatting_sample_title_authors_affiliations.pdf

Formatting changes have been made where necessary, please see XYZ:

#2. Please note that funding information should not appear in any section or other areas of your manuscript. We will only publish funding information present in the Funding Statement section of the online submission form. Please remove any funding-related text from the manuscript.

All funding statements have been removed, please see lines 463-468 on the tracked changes document.

3. Please upload a copy of Figure 6, to which you refer in your text on page 25. If the figure is no longer to be included as part of the submission please remove all reference to it within the text.

Apologies for this error. Figure 6 should have been Supplemental 2, please see changes in text.

5. We notice that your supplementary figures are uploaded with the file type 'Figure'. Please amend the file type to 'Supporting Information'. Please ensure that each Supporting Information file has a legend listed in the manuscript after the references list.

This error has been corrected.

#Reviewer 1:

The study design is interesting and align with the scope of PlosOne, I believe it can provide useful information; however, different sections of the manuscript need to be corrected to a large extent.d. I will outline below point by point some suggestions for improvement, but the main issues that need to be addressed are as follows:

• Results section: It is disorganized, data is repeatedly presented in both tables and figures, and the significance levels are not included in the tables and figures.

• Stated objectives: The necessary methodology to address all of the objectives has not been applied, and the objectives themselves are not properly written.

• Writing of the discussion.

Thank you for stating you found our research interesting and within the scope of PlosOne. The results section, objectives and discussion will all be addressed as per your following comments. Thank you for this feedback.

Point by point:

Why do the authors refer to it as "COD" throughout the text, but use "Turn" in the title? I understand that this may be to avoid repetition of keywords, but it might be more appropriate to switch them.

Thank you for highlighting this, as demonstrated in previous research (Dos’Santos, 2022), these terms are interchangeable and this is what we’ve done in this research too. To make this clearer we have added the following line to the first paragraph of the introduction:

These turning metrics (the term turning is synonymous with change of direction [COD]; 25; 51; 17) are often overlooked….

Why does the title include the term “significant”?

The term ‘significant turn’ refers to a change of direction with a deceleration³ <-2 m/s2, an angle change in direction of travel ≥20°, and a subsequent acceleration ≥2 m/s2, all within a 1 second duration. A significant turm has been referred to in literature previously (Dos’Santos, 2022) and is used to focus on change of direction movements that are believed to have a higher biomechanical load.

Introduction Section

In the introduction, some aspects need to be more thoroughly justified. The authors conclude by stating they will use Sportlight’s® LiDAR technology and provide only a justification for its use, but this approach is not appropriate for the introduction. Instead of saying what they will use, the authors should describe the technologies available for measuring this aspect, their characteristics, and what factors need to be considered. The introduction is meant to establish context and provide a theoretical framework, not to describe the specific methodology that the authors will use.

The following has been added to supply the reader with more context surrounding the rationale behind using LiDAR rather than any other tracking systems:

Tracking technology, from a sports science perspective, is used predominantly for external-load monitoring in an effort to reduce injury incidence rates and optimize performance (52). Daily use of this equipment allows for session-to-session adjustment of training periodisation to ensure that the training loads physically prepare players for the demands of a match (53; 54). The current industry-dominating tracking system in soccer is the global positioning system (GPS) (55). Since the in-competition legalization of GPS technology in 2015 by the International Football Association Board, it has been used by practitioners world-wide to provide on-pitch, external load metrics (55). It boasts benefits such as its relative affordability, its portable nature, and an instant feedback system; allowing for constant monitoring during both training and matches (53; 20). These GPS systems, when evaluated against the gold-standard 3D-motion capture system, yielded lower error values than other tracking systems such as video-based systems and local positioning systems (LPS)((20). However, with the recent integration of LiDAR technology, some of the current limitations faced by GPS, for example, satellite signals being blocked by stadiums/buildings, negative correlation between number of satellites signalling to the receiver and total distance and velocity measurement error, could lead to reduced usage in elite soccer. However, a validity study directly comparing LiDAR and GPS needs to be completed before any conclusions can be drawn from these assumptions.

Additionally, I believe it is necessary to delve deeper into the quantitative demands of soccer, specifically in relation to COD.

Currently I don’t think there is enough literature to ‘delve’ much deeper without waffling?? Unless I go generic not COD?

Line 55: This is the first time the acronym "COD" appears.

Full spelling has been written out.

“Change of direction (COD)”

Lines 65-67: Please use the acronym correctly.

The error lies in the abstract, where I have used the term “centre-backs”

Line 61: The terms "Turn" and "COD" seem to be used interchangeably. Are they meant to refer to the same thing?

Yes, they are. Please refer to earlier change made on line 57-58:

These turning metrics (the term turning is synonymous with change of direction [COD]; 25; 51; 17) are often overlooked….

Line 67: What is meant by “absence”?

The term ‘absence of turn characteristics’ is trying to highlight to the reader that the studies which are described in the previous sentences are stating only the number of turns without the additional context of what type of turns they are (which would be highlighted by turn characteristics). I feel this is described in the following sentences so there may not be need for a change in language on this occasion:

“The absence of turn characteristics is important because the biomechanical demands of these turns on the body are angle and velocity dependant (5,9–13). Therefore, quantifying the number of COD movements alone lacks scientific purpose without the added context of entry speed and turn angle. Both factors influence the braking and propulsive force characteristics……..”

Lines 93-100: The objectives need to be revised and not listed as bullet points. Additionally, the authors are not analysing the demands across different competitions, but rather within one team in the same competition. The last objective cannot be answered with the proposed methodology and is not addressed throughout the manuscript.

This feedback has been addressed and the following paragraph has replaced lines 93-100:

Building on the recent work of Dos’Santos et al. (17), this study aimed to use Sportlight’s® LiDAR technology to determine the turn demands experienced by soccer players in a professional team competing in the Premier League, UEFA Europa League, FA Cup and League Cup.. Specifically, the study sought to analyze turn characteristics for each playing position (goalkeeper, central defender, full-back, central midfielder, wide forward, and center forward) and explore the relationships between different turn characteristics. Examine differences in turn demands between competitions (Premier League, FA Cup, League Cup, and UEFA Europa League), for a single team. Finally, this research aimed to provide insights practitioners can use to inform injury-prevention, rehabilitation, and return-to-play programs, though further validation of the findings is necessary before generalizing to normative data for wider application.

Methods Section

Lines 108-131: Can the authors include values for the validity, reliability, and reproducibility of the methodology? The description currently only outlines the process but does not report its validity.

I’m not sure I can do this? Theo Bampouras did the reliability study on Sportlight but that isn’t a reliability study of my methods?

Lines 143-144: I do not understand the following statement: “If players had played in multiple positions, they were categorized into the position group in which they had played most matches.” Wouldn't it be more appropriate to categorize them based on the position they played in each match?

This is something that is a limitation of this research. Given the format in which the data was received (anonymized and with only each individual turn incident data), it was not possible to determine individual match data for positions. As stated in our study, no player had played less than 80% of their games in their allocated position group. Though this is not ideal, it is as close as we are able to get .

Line 144: The phrase "29 players played 80% of the matches" is unclear. Does this mean "No player had played less than 80% of their games in their allocated position group"?

This has been altered to the suggested phrasing:

No player had played less than 80% of their games in their allocated position group.

Line 146: Please include a reference to justify that 85 minutes is considered a full session. Previous studies have used fewer minutes.

I’ve searched through lots of different references and cannot find a reference. I used 85mins as this is what sportlight said they used, retrospectively I should have had some better rationale for this!

Definition of significant turns: This needs to be more clearly and precisely defined.

This is the current definition:

“Specifically, this was defined as a change of direction with a deceleration <-2 m/s2, an angle change in direction of travel ≥20°, and a subsequent acceleration ≥2 m/s2, all within a 1 second duration. Turns were further sub-categorised by angle and entry speed (Table 1) (17).”

The author is unsure how the reviewer would like this to be more precise. These are the only metrics we use to define a significant turn and it is the definition that has been used in previous literature.

Statistical analysis: Please explain the process of determining the interquartile range.

This information has been added in the penultimate sentence of the statistical analysis section:

“The interquartile range (IQR) was calculated by subtracting the first quartile (25th percentile) from the third quartile (75th percentile) of the dataset.”

Results:

Table 2 Title: There is an unnecessary period.

This error has been fixed.

Table 2: I believe the table could be formatted differently to improve comprehension. Instead of listing data sequentially, perhaps several values could be included within the same cell (in parentheses)? Why report both the mean and the median?

A new table has been inserted. Here are the changes which have been made:

1. Consolidation of Metrics: Each cell now contains the mean ± SD followed by the IQR in parentheses for turn characteristics. For total turns, it includes the mean ± SD followed by the range (min–max).

2. Removal of Medians: As the reviewer questioned their necessity, they’ve been removed for clarity and brevity.

3. Improved Layout: Grouped data by positions and added column headings for better readability.

Position Low Angle (Mean ± SD, IQR) Medium Angle (Mean ± SD, IQR) High Angle (Mean ± SD, IQR) Total Turns (Mean ± SD, Min–Max)

GK 1.44 ± 0.73 (1.00) 1.57 ± 1.16 (1.00) 3.59 ± 2.47 (2.00) 4.77 ± 3.32 (1–15)

FB 2.88 ± 1.69 (2.00) 6.05 ± 3.50 (4.75) 18.10 ± 7.65 (10.00) 26.36 ± 10.32 (2–51)

CD 3.03 ± 1.76 (2.00) 5.47 ± 2.50 (3.00) 15.03 ± 8.31 (6.00) 22.82 ± 9.41 (1–54)

CM 3.38 ± 1.96 (2.00) 7.38 ± 3.48 (5.00) 23.86 ± 15.72 (11.00) 34.52 ± 17.93 (3–102)

WF 2.48 ± 1.56 (2.00) 4.40 ± 2.36 (3.00) 14.38 ± 8.98 (8.75) 20.70 ± 11.18 (2–68)

CF 1.71 ± 0.95 (1.50) 7.00 ± 1.15 (1.50) 20.57 ± 3.51 (5.50) 29.83 ± 4.49 (25–37)

Overall 2.92 ± 1.79 5.82 ± 3.35 16.84 ± 12.14 24.50

Line 124: Should this reference be to Table 2 or Table 3?

Unsure which line this is referring to as there isn’t a reference to Table 2 or 3, however, all references have been checked and confirmed correct.

What is the difference between Table 1 and Figure 1, apart from expressing the same data in different formats? In my opinion, this is redundant. Perhaps only the figures should be included. Please also consider my earlier comments regarding Table 2.

Is this supposed to be referring to Figure 1 and Table 2? If so, we thank the reviewer for their opinion on this, however, we feel Table 2 summarises all data from both Figure 1, 2 together and therefore is of benefit to the reader. In addition, given the large axis range, we feel the figure should be used as a visual demonstration of the spread of data, whereas the table provides accurate absolute data detailing the number of turns in each turn characteristic group that wouldn’t otherwise be accurately deduced from the figure.

Line 203: The authors state “No significant differences were found between outfield players (p > 0.05) (Fig 2d),” but then report differences between positions.

Apologies, this is a mistake as it was supposed to be referring to just low-angled turns. This has now been corrected:

No significant differences were found between outfield players for low angled turns (p > 0.05) (Fig 2d).

The Results section needs a complete revision.

We apologise to the reviewer for the unclear results section. We have separated and addressed the rest of this comment:

It is not well organized, as the authors first present all the tables and figures, then repeat each one.

We have removed the summary paragraph at the beginning of the results section whereby all figures and tables were initially introduced.

Additionally, figure captions appear within the text, the narrative jumps from one table to another, and then back again.

All figure captions have been made to be clearly not within text by adding a textbox around each caption. Additionally, all references to Table 2 are before all references to Table 3.

Lines 260-261: I believe the heading is not appropriate for the content that follows.

We understand your issues with this heading and have changed it to:

“Investigating how entry speed changes with turn angle”

Why are there no indicators (simbols) in the tables and figures to help interpret significant differences?

This was initially involved in the table, however, due to the number of significant differences there are between all of the position groups, the authors felt this caused more confusion rather than information. If this is something which is required it can be included.

Discussion Section:

Lines 308-318: The writing and composition of the first paragraph need improvement. Follow the order of the objectives when reporting the main findings. It is not correct to state, “The aim was to build upon research by Dos’Santos et al. (17)”—the objectives are different. Avoid referring to tables and figures

---

## [Decision Letter · Decision Letter 1]

21 Jan 2025

PONE-D-24-31277R1An Exploratory Analysis Investigating the Turn Demands of the Premier League, FA Cup, League Cup and UEFA Europa League for an English Premier League Soccer TeamPLOS ONE

Dear Dr. Griffiths,

Thank you for submitting your manuscript to PLOS ONE. After careful consideration, we feel that it has merit but does not fully meet PLOS ONE’s publication criteria as it currently stands. Therefore, we invite you to submit a revised version of the manuscript that addresses the points raised during the review process.

Please consider the minor suggestions from reviewer 1.

We look forward to receiving your revised manuscript.

Kind regards,

Filipe Manuel Clemente, PhD

Academic Editor

PLOS ONE

Journal Requirements:

Reviewers' comments:

Reviewer's Responses to Questions

**Comments to the Author**

1. If the authors have adequately addressed your comments raised in a previous round of review and you feel that this manuscript is now acceptable for publication, you may indicate that here to bypass the “Comments to the Author” section, enter your conflict of interest statement in the “Confidential to Editor” section, and submit your "Accept" recommendation.

Reviewer #1: All comments have been addressed

Reviewer #2: All comments have been addressed

2. Is the manuscript technically sound, and do the data support the conclusions?

Reviewer #1: Yes

Reviewer #2: Yes

3. Has the statistical analysis been performed appropriately and rigorously? 

Reviewer #1: Yes

Reviewer #2: Yes

4. Have the authors made all data underlying the findings in their manuscript fully available?

Reviewer #1: Yes

Reviewer #2: Yes

5. Is the manuscript presented in an intelligible fashion and written in standard English?

Reviewer #1: Yes

Reviewer #2: Yes

6. Review Comments to the Author

Reviewer #1: Dear Authors,

I would like to acknowledge the effort you have made, which has undoubtedly improved the readability and comprehension of the manuscript, particularly in the Results section. The data are no longer duplicated and now follow a logical order, which enhances clarity. I have, however, a few minor considerations:

1. Introduction Section

While the authors describe the match demands in women’s soccer, why is there no mention of the demands in men’s soccer, given that this study was conducted with a male sample?

2. First Paragraph of the Discussion

I believe the first paragraph of this section should be restructured to follow the conventional approach for discussions. This means emphasizing the study’s objective and highlighting the main findings without referring to tables or supplementary materials. Please also review the first sentence of the following paragraph for clarity and relevance.

3. Competitions Categorization

Since the authors did not analyze turning movements separately for each competition but instead grouped regular season and knockout competitions together, should this not be explicitly stated throughout the manuscript (title and objectives for example)? There is potential for confusion when the title and objective suggest a distinction, but the data are not presented separately for these competitions.

4. Table Numbering

Is it correct for the current Table 4 to retain this numbering, or should it be Table 2 due to the inclusion of supplementary material? Please verify and ensure consistency.

5. GPS Accuracy

In the Introduction, the authors claim that GPS devices have a lower error margin than other tracking systems, such as LPS. Is this accurate? Could the authors confirm this with additional references to strengthen the statement?

I hope these considerations assist in further refining your manuscript.

Reviewer #2: (No Response)

7. PLOS authors have the option to publish the peer review history of their article (what does this mean? ). If published, this will include your full peer review and any attached files.

**Do you want your identity to be public for this peer review?** For information about this choice, including consent withdrawal, please see our Privacy Policy .

Reviewer #1: No

Reviewer #2: No

---

## [Author Response · Author response to Decision Letter 1]

5 Mar 2025

Rebuttal document for: ‘Exploring Turn Demands of an English Premier League Team Across League and Knockout Competitions Over a Full Season’ by Griffiths et al.

We would like to thank the reviewers for their comments on our work, which provided suggestions that have further improved the manuscript. We have responded to each of the reviewer comments below and where we have made a careful revision of the manuscript, these changes are tracked changes. If you have any questions regarding our rebuttal or revisions, please do not hesitate to contact us.

Reviewer #1:

Dear Authors,

I would like to acknowledge the effort you have made, which has undoubtedly improved the readability and comprehension of the manuscript, particularly in the Results section. The data are no longer duplicated and now follow a logical order, which enhances clarity. I have, however, a few minor considerations:

The authors thank the reviewer for taking their time to comment on and improve this manuscript.

1. Introduction Section

While the authors describe the match demands in women’s soccer, why is there no mention of the demands in men’s soccer, given that this study was conducted with a male sample?

The authors thank the reviewer for pointing out this error. This has been corrected and metrics from the same systematic review has been included but for male soccer players:

Line 55-56:

Linear demands such as high-speed and sprint running distances ranged from 618 to 1,001 m and 153–295 m, respectively, in professional male soccer players (64).

I believe the first paragraph of this section should be restructured to follow the conventional approach for discussions. This means emphasizing the study’s objective and highlighting the main findings without referring to tables or supplementary materials. Please also review the first sentence of the following paragraph for clarity and relevance.

We have redrafted the first paragraph of the discussion to rearticulate the objective of the study and present the key findings. We have done this without referring to tables or figures as advised. We have also redrafted the first sentence of the next paragraph to add clarity.

Please see re-drafted paragraph below (line 305-318):

The present study sought to determine the turn demands of soccer players in each playing position using data from the English Premier League, FA Cup, League Cup and Europa League of a single team. The primary findings showed no difference in turn demands between competition types (Premier League vs. knockout matches), but significantly greater total turns performed by CM (~35) compared to GK (~5), CD (~23) and WF (~21), and differences in turn characteristics between each position group for each angle and entry speed category.

In contrast to the present study’s findings, similar research from Dos’Santos et al. (17), showed CMs performed more turns on average (~38) compared to other positional groups (FB, CD, CF). Though CM performed more turns, the rationale for CM performing the most turns in both studies was shared; this was attributed to their duel attacking and defending roles.

3. Competitions Categorization

Since the authors did not analyze turning movements separately for each competition but instead grouped regular season and knockout competitions together, should this not be explicitly stated throughout the manuscript (title and objectives for example)? There is potential for confusion when the title and objective suggest a distinction, but the data are not presented separately for these competitions.

The authors thank the reviewer for this comment. We have acted upon the advice given and clarified this throughout the manuscript. Please see below the changes made to the title and within the manuscript:

Title:

‘An Exploratory Analysis Investigating the Turn Demands of the Premier League, FA Cup, League Cup and UEFA Europa League for an English Premier League Soccer Team’

‘Exploring Turn Demands of an English Premier League Team Across League and Knockout Competitions Over a Full Season’

Line 123-126:

Turning data were obtained from 49 match fixtures during 2022-23 season, from the Premier League (35 matches), UEFA Europa League (5 matches), League Cup (5 matches) and FA Cup (4 matches), the latter three competitions are grouped for analysis as ‘knockout’ competitions.

Line 308:

The primary findings showed no difference in turn demands between competition types (Premier League vs. knockout matches),

Line 420-423:

Despite these previous findings, the current study found no significant differences between turn characteristics when comparing knockout (FA Cup, League Cup and UEFA Europa League) vs. league soccer (Premier League).

4. Table Numbering

Is it correct for the current Table 4 to retain this numbering, or should it be Table 2 due to the inclusion of supplementary material? Please verify and ensure consistency.

The authors thank the reviewer for identifying this error. Line 299 now shows the table to be named, Table 2.

5. GPS Accuracy

In the Introduction, the authors claim that GPS devices have a lower error margin than other tracking systems, such as LPS. Is this accurate? Could the authors confirm this with additional references to strengthen the statement?

The authors have re-worded the following to provide a clearer and more accurate narrative for the reader:

Line 96-99:

These GPS systems, when evaluated against the gold-standard 3D-motion capture system, yielded lower error values than than other tracking systems such as video-based systems and local positioning systems (LPS), but lower validity than local positioning systems (LPS)(64), as well as often failing to quantify turn metrics (20).

---

## [Editor Report · Decision Letter 2]

7 Mar 2025

Exploring Turn Demands of an English Premier League Team Across League and Knockout Competitions Over a Full Season

PONE-D-24-31277R2

Dear Dr. Griffiths,

We’re pleased to inform you that your manuscript has been judged scientifically suitable for publication and will be formally accepted for publication once it meets all outstanding technical requirements.

Kind regards,

Filipe Manuel Clemente, PhD

Academic Editor

PLOS ONE
---

## [Editor Report · Acceptance letter]

PONE-D-24-31277R2

PLOS ONE

Dear Dr. Griffiths,

I'm pleased to inform you that your manuscript has been deemed suitable for publication in PLOS ONE. Congratulations! Your manuscript is now being handed over to our production team.

Kind regards,

on behalf of

Dr. Filipe Manuel Clemente

Academic Editor

PLOS ONE